# The Genus *Miconia* Ruiz & Pav. (Melastomataceae): Ethnomedicinal Uses, Pharmacology, and Phytochemistry

**DOI:** 10.3390/molecules27134132

**Published:** 2022-06-27

**Authors:** Viviane Bezerra da Silva, José Weverton Almeida-Bezerra, Adrielle Rodrigues Costa, Maria Flaviana Bezerra Morais-Braga, Maraiza Gregorio de Oliveira, Anderson Angel Vieira Pinheiro, Raimundo Samuel Leite Sampaio, José Walber Gonçalves Castro, Marcos Aurélio Figueiredo dos Santos, Valdilia Ribeiro de Alencar Ulisses, Maria Edilania da Silva Serafim Pereira, Dieferson Leandro de Souza, Bruno Melo de Alcântara, Maria Elizete Machado Generino, José Thyálisson da Costa Silva, Ademar Maia Filho, Sabrina Bezerra da Silva, Myunghan Moon, Bonglee Kim, José Galberto Martins da Costa

**Affiliations:** 1Department of Botany, Federal University of Pernambuco (UFPE), Av. Prof. Moraes Rego, 1235, Recife 50670-901, Brazil; viviane.silva@urca.br; 2Laboratory of Applied Mycology of Cariri (LMAC), Regional University of Cariri (URCA), Av. Cel Antônio Luis, 1161, Pimenta, Crato 63105-010, Brazil; adrielle.arc@hotmail.com (A.R.C.); flavianamoraisb@yahoo.com.br (M.F.B.M.-B.); 3Department of Biological Sciences, Federal University of Cariri (UFCA), Av. Ten. Raimundo Rocha, 1639-Cidade Universitária, Juazeiro do Norte 63048-080, Brazil; maraaiza0104@hotmail.com; 4Institute for Research in Drugs and Medicines—IpeFarM, Federal University of Paraíba, Cidade Universitária, João Pessoa 58051-970, Brazil; andersonangelvieira@gmail.com; 5Department of Biological Sciences, Universidade Regional do Cariri (URCA), Av. Cel Antônio Luis, 1161, Pimenta, Crato 63105-010, Brazil; samsampaio@hotmail.com (R.S.L.S.); walbercastro1@hotmail.com (J.W.G.C.); marcos.figueiredo@urca.br (M.A.F.d.S.); valdilia_rau@yahoo.com.br (V.R.d.A.U.); mserafimedilania@gmail.com (M.E.d.S.S.P.); diefersonleandro@gmail.com (D.L.d.S.); brunomelo870@gmail.com (B.M.d.A.); maria.machado@urca.br (M.E.M.G.); jose.thyalisson@urca.br (J.T.d.C.S.); ademarfilho_9@hotmail.com (A.M.F.); sabrina.silva@urca.br (S.B.d.S.); 6Department of Pathology, College of Korean Medicine, Kyung Hee University, Seoul 02447, Korea; audgksdl5364@khu.ac.kr; 7Laboratory of Natural Products Research, Regional University of Cariri (URCA), Crato 63122-290, Brazil; galberto.martins@gmail.com

**Keywords:** bioactivity, ethnobotany, ethnopharmacology, medicinal plants, natural product

## Abstract

Species of the genus *Miconia* are used in traditional medicine for the treatment of diseases, such as pain, throat infections, fever, and cold, and they used as depuratives, diuretics, and sedatives. This work reviewed studies carried out with *Miconia* species, highlighting its ethnomedicinal uses and pharmacological and phytochemical potential. This information was collected in the main platforms of scientific research (PubMed, Scopus, and Web of Science). Our findings show that some of the traditional uses of *Miconia* are corroborated by biological and/or pharmacological assays, which demonstrated, among other properties, anti-inflammatory, analgesic, antimutagenic, antiparasitic, antioxidant, cytotoxic, and antimicrobial activities. A total of 148 chemical compounds were identified in *Miconia* species, with phenolic compounds being the main constituents found in the species of this genus. Such phytochemical investigations have demonstrated the potential of species belonging to this genus as a source of bioactive substances, thus reinforcing their medicinal and pharmacological importance.

## 1. Introduction

Melastomataceae is one of the largest families of angiosperms, represented by about 5750 species distributed in 177 genera [1,2], occurring in tropical and subtropical regions around the world [3]. In Brazil, Melastomataceae is the fifth largest family, in terms of diversity of species, with *Miconia* Ruiz & Pav., *Leandra* Raddi, and *Tibouchina* Aubl. being the most representative genera [4,5].

*Miconia* has round 1900 species; in Brazil, 289 species of this genus are found, among which, 42% are endemic, distributed in different phytogeographic domains (Amazon, Caatinga, Cerrado, and Atlantic Forest) [6]. Members of this genus are used in the treatment of health conditions such as pain [7], throat conditions [8], and fever and cold symptoms [9,10], as well as a depurative, diuretic, and sedative [11]; some studies have already cited its popular uses [12,13].

*Miconia* species have been frequently reported in the literature because of their diverse biological and pharmacological properties, including anti-inflammatory [12,13,14] and analgesic actions [15], antiparasitic [16,17,18], antimicrobial [19,20], cytotoxic [21,22], antimutagenic [23], phytotoxic [24,25], and insecticidal activity [26].

According to phytochemical analyzes of *Miconia* species, chemical compounds belonging to the classes of triterpenes, flavonoids, phenolic acids, and steroids were identified, and the findings of these researches demonstrated the biological and pharmacological potential of some of them [17,22,27,28]. Viegas et al. [18] reported that the compound primin, isolated from *M. willdenowii*, showed promising leishmanicidal action. Silva et al. [22] showed that the flavonoid Matteucinol, isolated from *M. chamissois,* showed cytotoxic action in vitro and in vivo against cancer cell lines (e.g., glioblastoma).

Given the importance of the genus *Miconia*, as well as the previous aspects mentioned, this research aimed to carry out a systematic review of the literature published in the last twenty-one years (2000–2021) on the traditional uses and biological, pharmacological, and phytochemical activities of the genus, thus assisting in the diagnosis of studies of these species and possibly identifying the existence of relevant patterns for these researches and providing directions for future bioprospecting studies.

## 2. Review

### 2.1. Ethnopharmacological Uses

Ethnomedicinal studies have reported different uses of *Miconia* species in folk medicine—for example, *M. albicans*, which is one of the most studied species of the genus. Known as “canela-de-velho”, the species is traditionally used by the Brazilian populations for the treatment of different diseases. The leaves of *M. albicans* are used in the preparation of teas, mainly consumed for the treatment of arthritic and back pain [7]. The use of the its stem to treat vitiligo and relieve symptoms of fever are also reported [10].

Based on a survey in the rural communities located in the state of Minas Gerais (Brazil), *M. rubiginosa*, locally known as “capiroroquinha”, was reported to have its leaves used in the treatment of throat conditions [8]. Boscolo and Valle [9] reported the use of *M. cinnamomifolia* leaves in the treatment of fever and cold by residents in the state of Rio de Janeiro. *Miconia chartacea* was reported by herbalists in the state of Rio de Janeiro for its ethnomedicinal indications [29].

### 2.2. Biological and Pharmacological Activities

The biological and pharmacological potential of *Miconia* has been extensively investigated. The most frequently reported activities are shown in Figure 1. Studies show that the antimicrobial potential was the most reported in the last 21 years, corresponding to 26% of the citations.

#### 2.2.1. Anti-Inflammatory and Analgesic Activities

The species of the genus *Miconia* have a great range of uses. Some are used in Brazilian folk medicine for their analgesic and anti-inflammatory properties, which increases the need to deeper investigations about these reported effects [14]. Spessoto et al. [30] evaluated the analgesic activity of hexane, methylene chloride, and ethanol extracts from *M. rubiginosa* leaves, which showed central and peripheral analgesic activity in mice and rats, through induced writhing by acetic acid and hot plate tests. According to these authors, this activity was probably mediated by the inhibition of prostaglandin synthesis and central inhibitory mechanisms. Using a similar methodology, Vasconcelos et al. [31] investigated the analgesic actions of the extracts (hexane, methylene chloride, and ethanol) from *M. albicans* leaves, which showed significant antinociceptive activities in the writhing test.

Vasconcelos et al. [15] investigated the analgesic action of olenoic (**131**) and ursolic (**135**) acids, compounds isolated from the methylene chloride extract of *M. albicans* leaves; they found that these substances inhibited abdominal constriction in mice induced by acetic acid.

Gatis-Carrazzoni et al. [14] investigated the antinociceptive and anti-inflammatory actions of the methanolic extract of the leaves of *M. minutiflora* on induced edema in the paw of rats and found a decrease in cell migration in experimental models; this was associated with the inhibition of the pro-inflammatory cytokines TNF-α (tumor necrosis factor-alpha) and IL-1β (interleukin 1β). Additionally, the compound acted on inflammatory pain and as agonists of α2-adrenergic receptors.

In the study by Lima et al. [12], the dry extract of *M. albicans* leaves showed anti-inflammatory activity in the edema induced by carrageenan injection in mouse paws. Oral treatment with the extract significantly decreased TNF-α and IL-1β levels, consequently reducing the inflammatory nociception. This corroborated with the results of Quintans-Junior et al. [32], which reported that the ethanolic extract of *M. albicans* significantly reduced the migration of leukocytes and levels of TNF-α and IL-1β. The extract reduced pain in mice, without any apparent damage to the liver, decreasing nociceptive and hyperalgesic behaviors and mechanical hyperalgesia.

According to Correa et al. [13], the methanolic extract of *M. minutiflora* leaves showed anti-inflammatory effects, thus inhibiting edema and the recruitment of leukocytes to the inflamed auricular tissue of mice. Gomes et al. [6] observed in their studies that *M. albicans* showed analgesic and anti-inflammatory effects in the treatment of patients with knee osteoarthritis, thus decreasing the pain of patients, an effect comparable to the pharmaceutical drug ibuprofen.

#### 2.2.2. Antimutagen

The evaluation of the extracts of four species of *Miconia* (*M. albicans*, *M. cabucu*, *M. rubiginosa*, and *M. stenostachya*) revealed antimutagenic effects from all extracts evaluated; in addition, none of the extracts at any of the concentrations tested (5, 10, and 20 μg/mL) showed mutagenic effects [23].

Gontijo et al. [16] demonstrated that the aqueous extract of the leaves of *M. latecrenata* leaves showed significant antimutagenic activity against strains of *Salmonella typhimurium*, which was observed using the Ames test; this aqueous extract showed a mutagenic inhibition of approximately 70% (340.0 μg/plate) and 61% (42.5 μg/plate) against the strains TA98 and TA97, respectively.

#### 2.2.3. Antiparasitic Activity

Cunha et al. [33] found that the triterpenes gypsogenic acid (**127**), oleanolic acid (**131**), and ursolic acid (**135**), isolated from *M. fallax* and *M. stenostachya*, were active against blood trypomastigote forms of *Trypanosoma cruzi*. In other investigations [34], these authors showed that the compounds oleanolic acid (**131**) and ursolic acid (**135**), isolated from two other species of *Miconia* (*M. sellowiana* and *M. ligustroides*), showed significant trypanocidal activity (*T. cruzi*) in vitro tests, with IC_50_ of 17.1 and 12.8 μM, respectively. Furthermore, the potassium salt derived from ursolic acid showed a great trypanocidal action (IC50 8.9 µM), when compared to the original compound. In an in vivo assay, the authors also demonstrated that ursolic acid (**135**) and its derivative showed a significant reduction of parasites at the parasitemia peak (75.7% and 70.4%, respectively).

According to Barrio et al. [35], *M. schlimii* had a trypanocidal effect on *T. cruzi* (≥75% at 500 mg/mL). Peixoto et al. [28] found that the crude hydroalcoholic extract of *M. langsdorffii* leaves and isolated compounds of oleanolic acid (**131**) and ursolic acid (**135**) showed moderate leishmanicidal activity against the promastigote forms of *Leishmania amazonenses*, with the IC_50_ values of 175.4 μg/mL, 360.3 mM, and 439.5 mM, respectively. *Miconia langsdorffii* also showed antiparasitic activity, where the crude extract and fractions of *n*-hexane and ethyl acetate were able to induce the death of adult *Schistosoma mansoni* worms.

Crude extracts from aerial parts of *M. langsdorffii* and some fractions of *n*-hexane and ethyl acetate were able to induce the death of *Schistosoma mansoni* worms [36]. In the studies by Lima et al. [37], the methanolic extract of *M. nervosa* leaves exhibited great in vitro antiplasmodial activity (IC_50_ = 9.9 ± 3.2 µg/mL) against *Plasmodium falciparum* strains.

According to Viegas et al. [17], the ethanolic extract of *M. willdenowii* leaves showed schistosomicidal activity, killing more than 50% of *Schistosoma mansoni* worms at all concentrations tested (25, 50, 75, and 200 μg/mL), when compared to praziquantel, which was used as the reference drug. In other investigations, the authors evaluated the leishmanicidal activity of *M. willdenowii*, where they observed that the ethanol extract and hexane fraction of the leaves of the species showed inhibitory activity against promastigote forms of *L. amazonensis*. The additional evaluation of metabolites isolated from *M. willdenowii* showed that the compound primin (**148**) had potent leishmanicidal action, with IC_50_ = 0.26 μg/mL (1.25 μM), a potency four times greater than the standard drug amphotericin B (IC_50_ = 5.08 μM) [18]. Gontijo et al. [16] demonstrated that aqueous and ethanolic extracts of *M. latecrenata* leaves had a great percentage of growth inhibition of *Plasmodium falciparum*.

#### 2.2.4. Antimicrobial Potential

Some species of *Miconia* stand out for having antimicrobial activities. For example, ethyl acetate extracts from the leaves of *M. lepidota* and two isolated benzoquinones (2-methoxy-6-heptyl-1,4-benzoquinone (**141**) and 2-methoxy-6-pentyl-1,4-benzoquinone (primin) (**148**)), showed antimicrobial activity against yeast strains of *Saccharomyces cerevisiae* [38].

The ethanolic extracts of three species of *Miconia* (*M. albicans*, *M. rubiginosa*, and *M. stenostachya*) exhibited an antimicrobial effect against some of the eleven microorganisms tested. The extracts of *M. albicans* and *M. rubiginosa* showed antibacterial effects against three gram-positive bacteria (*Staphylococcus aureus*, *S. saprophyticus*, and *Streptococcus agalactiae*) and two gram-negative bacteria (*Shigella flexneri* and *Klebsiella pneumoniae*). While the *M. stenostachya* extract was only active against *C. albicans* [39].

Garcia et al. [40] observed the antibacterial activity of methanolic extracts from *M. mexicana* leaves, showing an MIC of 0.5 μg/mL against *S. aureus*. According to Cunha et al. [41], triterpenic acids (gypsogenic (**127**), oleanolic (**131**), sumaresinolic (**134**), and ursolic (**135**) acids isolated from *Miconia* species (*M. fallax*, *M. albicans*, and *M. stenostachya*) showed antimicrobial effects against six different bacterial strains (*Streptococcus mutans*, *S. mitis*, *S. sanguinis*, *S. salivarius*, *S. sobrinus*, and *Enterococcus faecalis*). Oleanolic (**131**) and ursolic (**135**) acids showed significant inhibitory effects, with MIC values ranging from 30 to 80 μg/mL.

Methanolic extracts from the leaves of *M. rubiginosa* and *M. stenostachya* and chloroform extracts from the leaves of *M. cabucu* showed antimicrobial activity against *Candida albicans* and eight different species of bacteria (*Bacillus subtilis*, *B. cereus*, *Staphylococcus epidermidis*, *S. aureus*, *Salmonella* spp., *E. faecalis*, *Escherichia coli*, and *Micrococcus luteus*), as tested using the disk diffusion method [42].

In the study from Pavan et al. [43], chloroform and methanol extracts from *M. cabucu* and *M. rubiginosa* showed inhibitory activity against strains of the bacteria *Mycobacterium tuberculosis*. Cunha et al. [44] demonstrated that the compounds isolated from *M. ligustroides* have inhibitory antibacterial activities. Oleanolic acid (**131**) showed inhibitory action against *B. cereus* and *Streptococcus pneumoniae*, with an MIC of 80 μg/mL for both bacteria. Ursolic acid (**135**) demonstrated antibacterial action against *B. cereus*, with an MIC of 20 μg/mL.

Bussmann et al. [45] reported that ethanolic and aqueous extracts of aerial parts of *M. salicifolia* showed antimicrobial potential against gram-negative (*E. coli*) and -positive (*S. aureus*) bacteria. According to Pinto et al. [46], the extract of *M. argyrophylla* leaves was promising in inhibiting mycelial growth and controlling the germination of *Colletotrichum lindemuthianum* conidia, in addition to being effective in reducing the severity of the disease—reducing it by 42%, compared to the control.

Using the broth microdilution method, Queiroz et al. [19] demonstrated that the ethanolic extract of the aerial parts of *M. rubiginosa* showed antimicrobial activity against five bacterial isolates, namely *E. faecalis*, *Kocuria rhizophila*, *E. coli*, *P. aeruginosa*, and *Salmonella choleraesuis*.

The aqueous extract from the leaves of *M. latecrenata* showed antibacterial activity against strains of *Staphylococcus aureus,* with an MIC value of 100.0 μg/mL [16]. Rodrigues et al. [47] also demonstrated that the extract prepared from the leaves of *M. latecrenata* was promising in inhibiting the bacterial growth of *S. aureus* and *Pseudomonas aeruginosa* strains; in addition, it demonstrated synergism with ampicillin and tetracycline, respectively.

In the study of Tomé et al. [20], the antimicrobial activity of the ethanol extract of *M. albicans* leaves against *Listeria innocua* was verified. The hexane fraction showed moderate inhibition against strains of *L. monocytogenes*, while the ethyl acetate fraction showed better inhibitory activity against *Listeria monocytogenes*, *L. innocua*, and *Bacillus cereus.*

Viegas et al. [18] revealed in their studies that the hexane fraction prepared from the leaves of *M. willdenowii* showed significant antimicrobial activity against strains of *S. aureus* and *Candida krusei*, showing IC_50_ values of 15.6 and 62.5 μg/mL, respectively. Compound isolated primin (**148**) also showed significant activity against *S. aureus* (IC_50_ = 8.94 μM), with similar results to the reference drug chloramphenicol (IC_50_ = 6.19 μM). Compound **148** also showed significant antifungal activity against all *Candida* sp. tested.

Gomes et al. [6] found that the aqueous extract of *M. chamissois* showed antimicrobial activity against *C. albicans* and *S. aureus*, having an MIC of 78.1 µg/mL and 312.5 µg/mL, respectively, but did not show activity against *E. coli.*

#### 2.2.5. Antioxidant Activity

Species of the genus *Miconia* have antioxidant properties evidenced in diverse studies, as demonstrated in the study of Boscolo et al. [48], where the authors found that *M. cinnamomifolia* had a moderate activity under the influence of these antioxidant properties of *Miconia*. Mancini et al. [49] confirmed the antioxidant activity of the flavonoids isolated from *M. alypifolia* leaves using ABTS (2,2′-Azinobis-(3-ethylbenzothiazoline-6-sulphonic acid). In the study of Mosquera et al. [50], the methanolic extract of *M. lehmannii* leaves showed significant antioxidant activity in the DPPH (2,2-Diphenyl-1-picrylhydrazyl) assays.

The methanolic extract, *n*-butanolic fraction, and flavonoids isolated from *M. albicans* demonstrated a great ability to scavenge AAPH (2,2′-azobis(2-amidinopropane) dihydrochloride) and DPPH radicals [51].

The methanolic extract of *M. albicans* fruits showed significant antioxidant activity [52]. Gontijo et al. [16] confirmed the antioxidant potential of the aqueous extract of *M. latecrenata* leaves in DPPH, β-carotene/linoleic acid, and lipid peroxidation assays, where the species showed important potential as a source of ROS (reactive oxygen species) inhibitors.

Lima et al. [12] demonstrated the antioxidant potential of the ethanolic extract of *M. albicans* leaves through in vitro antioxidant assays, including DPPH and ABTS free radical scavenging assays, ferric reducing antioxidant assay, NO scavenging assay, metal ion (Fe^2+^) chelating activity, and antioxidant capacity by inhibiting lipid peroxidation (TBARS). Correa et al. [13] also found the antioxidant activity of the methanolic extract of *M. albicans* fruits using DPPH and ABTS.

Gomes et al. [6], when evaluating through DPPH and phosphomolybdenum radical reduction methods, found the antioxidant activity of aqueous extracts of *M. chamissois* leaves, demonstrating an IC_50_ close to that observed for ascorbic acid.

#### 2.2.6. Cytotoxicity

According to Gunatilaka et al. [38], the ethyl acetate extract from the leaves of *M. lepidota* and isolated quinones 2-methoxy-6-heptyl-1,4-benzoquinone (**141**) and 2-methoxy-6-pentyl-1,4-benzoquinone (primin) (**148**) exhibited cytotoxic activity in murine cell lines.

Calderon et al. [53], when evaluating the in vitro ability of the extracts from two species of *Miconia* to inhibit cancer cells, observed that the extract of *M. impetiolaris* leaves showed low cytotoxicity, whereas the extract of the aerial parts of *M. serrulata* showed significant cytotoxicity, with a value of IC_50_ less than 12 µg/mL.

Serpeloni et al. [23] evaluated the cytotoxic activity of *Miconia* species in fibroblast cells of the lungs of Chinese hamster lung, reporting a cytotoxic effect on cells exposed to methanolic extracts of *M. albicans* and *M. cabucu* at a concentrations of 40 μg/mL, as well as *M. stenostachya* at 60 μg/mL. At lower concentrations of the extracts (5, 10, and 20 μg/mL), cytotoxicity was not observed.

The *M. minutiflora* extract showed low cytotoxicity against tumor cell lines using the Alamar blue assay [21]. Primin (**148**), a compound isolated from the leaves of *M. willdenowii*, revealed a cytotoxic potential against human peripheral blood mononuclear cells (CC_50_ = 53.12 μg/mL) [17].

Studies have shown that the compound Matteucinol (**38**), isolated from *M. chamissois*, has cytotoxic activity in human GBM (glioblastoma) cell lines, in both in vitro and in vivo, thus demonstrating great specificity [54]. Cunha et al. [55] evaluated the cytotoxic activity of *M. burchellii* extract and fractions using the MTT (3-(4,5-dimethyl-2-thiazolyl)-2,5-diphenyl-2-H-tetrazolium bromide) assay, where the ethyl acetate fraction showed potent cytotoxic action against four of the five of the tumor cell lines evaluated, while the ethyl ether fraction was cytotoxic only against the leukemia cell lines (IC_50_ 4.74 mg mL^−1^).

Rosa et al. [56], when evaluating the cytotoxicity of crude extracts of two species of *Miconia* (*M. albicans* and *M. chamissois*), found that the species showed IC_50_ values lower than 30 µg/mL against the cell lines of human cervical cancer.

According to Gomes et al. [6], the aqueous extracts of *M. chamissois* leaves showed low cytotoxicity on the cell lines evaluated, displaying IC_50_ values of de 182.7, 414.6, and 977.0 µg/mL for human keratinocytes, mouse fibroblasts, and mouse macrophages, respectively.

#### 2.2.7. Other Activities

Cunha et al. [57] showed that the ethanolic extract of *M. fallax* and mixture of triterpenoid acids (**131** and **135**) demonstrated tumor growth inhibition (HeLa cells). Serpeloni et al. [58], using the comet assay with extracts of *Miconia* (*M. albicans*, *M. cabucu*, *M. rubiginosa*, and *M. stenostachya*), found that all extracts evaluated showed low levels of genotoxicity.

Pinto et al. [24] found that the aqueous extract of *M. chamissois* leaves showed a phytotoxic effect on the germination and growth of *Cucumis sativus* (cucumber) seedlings. Santos et al. [25], when evaluating the phytotoxicity of the extracts obtained from seven species of *Miconia* (*M. albicans*, *M. alborufescens*, *M. ciliata*, *M. ibaguensis*, *M. lingustroides*, *M. minutiflora*, and *M. stenostachya*), observed that all extracts significantly inhibited the length of the radicles of *Lactuca sativa*. Furthermore, the *M. stenostachya* extract at 50% concentration considerably reduced the number of germinated seeds of the tested plant.

The ethanolic extract of *M. ferruginata* leaves had an insecticidal effect on *Spodoptera frugiperda* caterpillars. The authors showed that caterpillars fed with the extract at a concentration of 1000 mg mL^−1^ had a mortality of 56.67%, when compared to the control, in addition to displaying the elongations of the pupal and larval stages, causing an effect during the early stages of the development of the caterpillar [26].

Vagas et al. [59] demonstrated that the aqueous extracts of *M. albicans* showed phytotoxic activity, thus reducing the length of the aerial part of lettuce seedlings, when increasing the concentrations of the extracts.

### 2.3. Phytochemistry

A total of 148 chemical compounds were identified in the *Miconia* species. Among them were alkaloids (**1**–**9**), flavonoids (**10**–**68**), phenolic acids (**69**–**101**), terpenoids (**102**–**111**), triterpenoids and steroids (**112**–**140**), and other compounds (**141**–**148**). Flavonoids and phenolic acids were the main components present in the genus. These classes of the compounds are directly or indirectly related to the pharmacological activities of *Miconia*. Furthermore, it was observed that the phytochemical and pharmacological studies of the genus have focused on the species *M. albicans*, *M. latecrenata*, and *M. ferruginata*. The names, species, and plant organs found, as well as the biological activities of these compounds, are summarized in the Appendix A, and their structures are shown in the Appendix A.

## 3. Materials and Methods

### 3.1. Databases and Search Strategy

Publications were searched on PubMed, Scopus, and Web of Science. The keyword “*Miconia*” was associated with “traditional use”, “medicinal use”, “ethnobotany”, “ethnopharmacology”, “biological activity”, “pharmacology”, “natural product”, “phytochemistry”, “bioactive”, “toxicity”, and “cytotoxicity”; the data collected were published between 2000 to 2021.

### 3.2. Criteria for Inclusion and Exclusion of Articles

We included in this review only the publications that met the following criteria: scientific articles with information on different uses in folk medicine; biological and pharmacological activities and chemical composition of species belonging to *Miconia* published in the last 21 years (2000–2021).

Review articles, academic texts (monograph, dissertations, and theses), abstracts published in events, books, and documents, as well as works that were outside the databases, were not considered in this research.

### 3.3. Data sorting and Organization

From the preliminary search, 142 articles were identified, of which, 54 articles were removed due to duplication. A total of 88 articles were analyzed in full, being selected based on the information contained in the titles, abstracts, and full texts. Finally, 67 articles were included in this review, which provided data related to the traditional uses, chemical composition, and biological and pharmacological activities of the species of the genus *Miconia* (Figure 2).

The results of the review were grouped into the categories “Ethnopharmacological uses”, “Biological and pharmacological activities”, and “Phytochemistry”; in the latter, the chemical compounds were gathered in a table, and their chemical structures were represented using figures.

## 4. Conclusions

Few species of *Miconia* were recorded with therapeutic indications, with *M. albicans* being the most cited species. Most of these were included in investigations regarding chemical composition and/or biological activities, except for *M. chartacea*, which did not have any record with a scientific approach.

Although the genus *Miconia* is composed of about 1900 species, few of these have been investigated for their pharmacological and phytochemical properties. Studies of the biological and pharmacological activities have focused on some species of the genus, such as *M. albicans, M. latecrenata*, and *M. ferruginata.* Phytochemical investigations have demonstrated the potential of the species belonging to this genus as a source of bioactive substances, thus reinforcing the medicinal and pharmacological importance of *Miconia.*

## Figures and Tables

**Figure 1 molecules-27-04132-f001:**
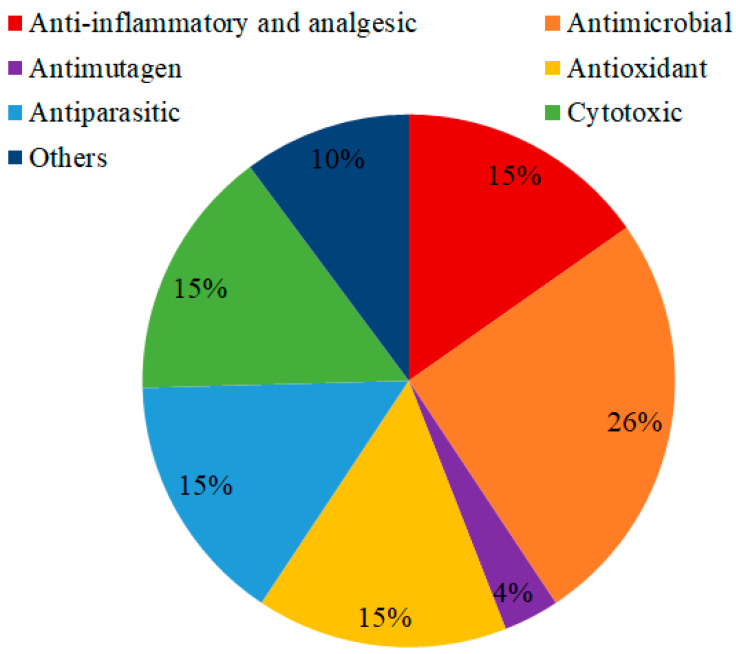
Biological and pharmacological activities reported for *Miconia* species in the last twenty-one years (2001–2021).

**Figure 2 molecules-27-04132-f002:**
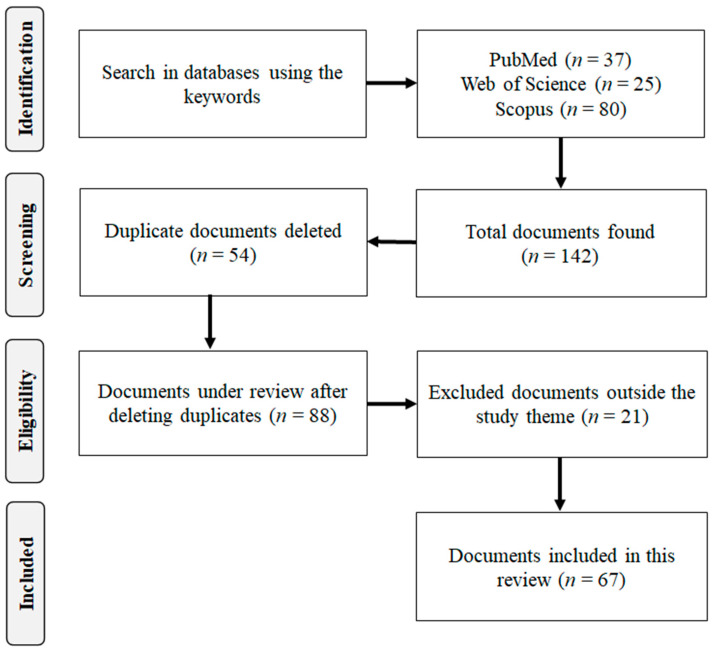
Selection flowchart of scientific documents included in this review.

## Data Availability

Not applicable.

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
