# Peer review of "The Genus Miconia Ruiz & Pav. (Melastomataceae): Ethnomedicinal Uses, Pharmacology, and Phytochemistry"

_molecules, 2022, doi:10.3390/molecules27134132_

Round 1

Reviewer 1 Report

The manuscript entitled -The genus Miconia Ruiz & Pav. (Melastomataceae): ethnomedicinal uses, pharmacology and phytochemistry- submitted by Viviane Bezerra da Silva et al. describes the molecules that could be obtained from the plants belonging to the genus Miconia.

The review is quite interesting and in a scatted literature try to put order with a molecular view. The methodology used and then the selection on the references is quite rigorous.

I suggest the publication to this journal after a major revision.

Table 1 and figure 2-7 are very long, so I suggest moving it to the supplementary material. Figure 2 -7 must use the same chemical format, sometime the hydrogens are explicit sometime are not, triple bond bent and so on, a careful revision of the structure is required.

For table 1 I also suggest adding the PubChem CID to easy retrieve all the basic information about the molecules.

In figure 1 put the biological and pharmacological activities close to the pie chart to improve the readability.

I strongly suggest adding a table/graph of all the plant reviewed grouped by the different genera, showing the molecules reported. Commenting on this table will guide the reader towards which species is most promising for future study, as suggested in lines 71-72.

Author Response

Reviewer 1: 

The manuscript entitled -The genus Miconia Ruiz & Pav. (Melastomataceae): ethnomedicinal uses, pharmacology and phytochemistry- submitted by Viviane Bezerra da Silva et al. describes the molecules that could be obtained from the plants belonging to the genus Miconia.

The review is quite interesting and in a scatted literature try to put order with a molecular view. The methodology used and then the selection on the references is quite rigorous.

I suggest the publication to this journal after a major revision.

Table 1 and figure 2-7 are very long, so I suggest moving it to the supplementary material. Figure 2 -7 must use the same chemical format, sometime the hydrogens are explicit sometime are not, triple bond bent and so on, a careful revision of the structure is required.

Dear Reviewer, as suggested Table 1 and Figure 2-7 have been moved to supplementary material. A careful review was made of the chemical format of the compounds.

For table 1 I also suggest adding the PubChem CID to easy retrieve all the basic information about the molecules.

Dear Reviewer, as suggested, the PubChem CID was added to the compounds in Table 1.

In figure 1 put the biological and pharmacological activities close to the pie chart to improve the readability.

Dear Reviewer, as suggested, the pie chart in Figure 1 has been edited to improve readability.

I strongly suggest adding a table/graph of all the plant reviewed grouped by the different genera, showing the molecules reported. Commenting on this table will guide the reader towards which species is most promising for future study, as suggested in lines 71-72.

Dear reviewer, we fully understand your request. Previously, we tried to add a table that emphasized the species, but that way it would be too long and the information repetitive, so we chose to set up a table that pointed out the compounds found in such species. In this way, we ask for understanding so that the analyzed plants continue in table 1, which also has the reported molecules.

Reviewer 2 Report

The MS entited "The genus Miconia Ruiz & Pav. (Melastomataceae): ethnome- dicinal uses, pharmacology and phytochemistry" was throughly checked and reviewed. The contanets are fine and i have given my suggestions in the MS. Here are some genra comments.

1. There are less potent phytochemicals in this genus  so why the authors choose to writeup?

2. the english language is poor and sentance structures should be modified.

3. Authors own work has not been reported. why? 

Author Response

The MS entited "The genus Miconia Ruiz & Pav. (Melastomataceae): ethnome- dicinal uses, pharmacology and phytochemistry" was throughly checked and reviewed. The contanets are fine and i have given my suggestions in the MS. Here are some genra comments.

  1. There are less potent phytochemicals in this genus so why the authors choose to writeup? Dear reviewer, considering the selection of references and the inclusion criteria, all articles containing the phytochemical analysis and the compounds present in Miconia species were added, also comprising the less potent compounds.
  2. the english language is poor and sentance structures should be modified. Dear reviewer, as requested, a thorough review of the English was carried out and the necessary changes were made.
  3. Authors own work has not been reported. why? Dear reviewer, I understand your doubt, but according to the methodology used and the inclusion criteria, such articles were not found in the search performed, therefore, they could not be added to the review.

Round 2

Reviewer 1 Report

The manuscript could be accepted in the current form

Reviewer 2 Report

Dear authors

The MS is accetable for me.